# A Fluidic Device for Immunomagnetic Separation of Foodborne Bacteria Using Self-Assembled Magnetic Nanoparticle Chains

**DOI:** 10.3390/mi9120624

**Published:** 2018-11-27

**Authors:** Gaozhe Cai, Siyuan Wang, Lingyan Zheng, Jianhan Lin

**Affiliations:** 1Key Laboratory of Agricultural Information Acquisition Technology, Ministry of Agriculture, China Agricultural University, Beijing 100083, China; gaozhe@cau.edu.cn; 2Key Laboratory of Modern Precision Agriculture System Integration Research, Ministry of Education, China Agricultural University, Beijing 100083, China; wangsiyuan@cau.edu.cn (S.W.); lingyanzheng@cau.edu.cn (L.Z.)

**Keywords:** fluidic chip, magnetic nanoparticle chains, immunomagnetic separation, *salmonella*

## Abstract

Immunomagnetic separation has been widely used for the separation and concentration of foodborne pathogens from complex food samples, however it can only handle a small volume of samples. In this paper, we presented a novel fluidic device for the specific and efficient separation and concentration of *salmonella*
*typhimurium* using self-assembled magnetic nanoparticle chains. The laminated sawtooth-shaped iron foils were first mounted in the 3D-printed matrix and magnetized by a strong magnet to generate dot-array high gradient magnetic fields in the fluidic channel, which was simulated using COMSOL (5.3a, Burlington, MA, USA). Then, magnetic nanoparticles with a diameter of 150 nm, which were modified with the anti-*salmonella* polyclonal antibodies, were injected into the channel, and the magnetic nanoparticle chains were vertically formed at the dots and verified using a fluorescence inverted microscope. Finally, the bacterial sample was continuous-flow injected, and the target bacteria could be captured by the antibodies on the chains, followed by gold standard culture plating to determine the amount of the target bacteria. Under the optimal conditions, the target bacteria could be separated with a separation efficiency of 80% in 45 min. This fluidic device could be further improved using thinner sawtooth-shaped iron foils and stronger magnets to obtain a better dot-array magnetic field with larger magnetic intensity and denser dot distribution, and has the potential to be integrated with the existing biological assays for rapid and sensitive detection of foodborne bacteria.

## 1. Introduction

Since foodborne diseases caused by pathogenic bacteria have posed a great threat to global public health in recent years, it has been essential to explore economical and efficient techniques for the screening of foodborne pathogens [1]. The procedure for bacteria detection generally includes sample collection and pre-enrichment, bacteria separation, and bacteria detection. Due to the very low concentration of bacteria in real food samples for routine screening, effective and rapid methods for the separation of target bacteria could allow bacterial analysis independent of sample sizes and volumes, and improve detection sensitivity as well [2]. 

Currently available bacteria separation methods mainly include filtration [3,4], centrifugation [5,6,7], and immunomagnetic separation (IMS) [8,9]. The common weakness of size-based filtration and mass-based centrifugation is that neither are able to specifically separate the target bacterial cells from those cells with similar physical characteristics in complex sample matrices. Different from filtration and centrifugation, IMS can specifically separate target bacterial cells through antigen-antibody reaction and has become very popular in practical applications for separation and enrichment of pathogenic bacteria. IMS has been reported for the detection of pathogens from various complex food matrices, such as meat [10], eggs [11], or fruits [12,13]. It first uses immune magnetic particles to capture target bacteria to form magnetic bacteria, then applies an external magnetic field to capture the magnetic bacteria and remove the sample background, and finally re-suspend the magnetic bacteria in a small volume of buffer solution to obtain purified and enriched target bacterial cells. IMS has usually been combined with polymerase chain reaction (PCR) [14,15,16], enzyme-linked immunosorbent assay (ELISA) [17,18], and lateral flow chromatography [19,20] to increase their detection sensitivity and shorten their detection time. However, conventional IMS often requires well-trained technicians due to the complex procedure and is not suitable to handle a large volume of samples due to the narrow working range of the magnetic field and large amount of magnetic nanoparticles.

With the fast development of microfluidics and microfabrication in the past decade, many efforts have been made to combine IMS with microfluidics and microfabrication to develop miniaturized, automatic, and rapid bioseparation methods. Magnetophoresis is an emerging bioseparation method combining IMS with microfluidics. A typical magnetophoretic bioseparation system often consists of a Y-typed channel and a magnetic field. A mixture of magnetic and non-magnetic substance is squeezed by a sheath flow to one side of the channel (mixture side), and will basically not shift to the other side (sheath flow side) at the absence of the magnetic field due to the laminar flow regime in the channel. When the magnetic field is applied at the sheath flow side, the magnetic substance will be attracted by the magnetic field and move towards the sheath flow side while the non-magnetic substance remains in the sheath flow, leading to magnetophoretic separation of the magnetic substance from the non-magnetic one. An interesting study on magnetophoretic devices was reported by Lee et al. [21] for bacteria separation from blood in a microfluidic chip, and a high separation efficiency of 100% was achieved at the flow rate of 60 mL/h. Magnetophoretic bioseparation has often been used for DNA extraction as well. Karle et al. [22] proposed a phase-transfer magnetophoretic method for continuous extraction of DNA. A rotating permanent magnet was applied to generate a time-varying magnetic field beside the microfluidic channel for continuous separation of the DNA captured by the magnetic beads. However, so far, most magnetophoretic separation systems could achieve the separation of the targets only after the targets were captured by the magnetic particles. Magnetic flow bioseparation is another new method combining IMS with microfluidics. A remarkable study on magnetic flow bioseparation was proposed by Lee et al. [23] for rapid separation and detection of pathogenic bacteria in a large volume of food samples using a 3D-printed cylindrical channel. This device was demonstrated to be able to separate and detect the target bacteria in 3 min with a low detection limit of 10 CFU/mL. Although it could achieve the capture and separation of the target bacteria, a very large amount of the immune magnetic nanoparticles (up to 200 μg, 10 times more than routine dose) was used to ensure efficient immunoreaction, resulting in very high cost, and this might greatly limit its practical application.

In recent years, many efforts have been made to enhance the separation efficiency of target bacteria avoiding the use of a large amount of magnetic nanoparticles. The forming of immune magnetic nanoparticle chains in microfluidic channels has been reported with the potential to enhance the separation efficiency [24,25], since the antibodies on the upright chains have more chances to react with the targets when they flow through the channel. When an external magnetic field was applied on magnetic particles, dipolar‒dipolar interactions could be induced between the particles [26]. As each magnetic particle was magnetized acting like a single magnet, they would combine with each other to form chains along the direction of the magnetic gradient. A common way to form magnetic particle chains in the microfluidic channel has been to place a permanent magnet at each side of the microfluidic channel with a special angle to form a high gradient magnetic field in the channel [27,28], and the magnetic particles could be magnetized and formed into chains when they were present in the channel. Recently, some new methods were reported to form magnetic particle chains in microfluidic channels. Saliba et al. [29] reported an interesting method to form self-assembled magnetic particle chains for sorting cancer cells using water-based ferrofluid as magnetic ink to print a magnetic trap array in the microfluidic channel, which could induce the magnetic particles to form the chains. By using the magnetic particle chains, a high separation efficiency (>94%) of B lymphoid cells from the mixture containing T and B lymphoid cells was achieved. Armbrecht et al. [30] developed a soft-magnetic lattice with a spacing of 350 μm and permanent magnets to form magnetic particle chains and it was successfully embedded in a microfluidic biosensing platform for detection of biotinylated glucose oxidase with a limit of detection (LOD) of 8 ng/mL. A magnetic fluidized bed, as a special case for magnetic particle chains, has also been reported with a higher density of magnetic particles to improve separation efficiency [31,32,33,34,35]. It was often formed by applying an external magnetic field and a constant flow in a cone-shaped microfluidic chamber, resulting in a steady-state dynamic regime to form the magnetic fluidized bed. Pereiro et al. [33] developed a magnetic fluidized bed in a microfluidic chip to realize a separation efficiency of 84–93% for *salmonella typhimurium* at concentrations from 10^2^ to 10^4^ CFU/mL. Since the concentration of many foodborne pathogenic bacteria in real food samples is very low, efficient bacteria separation methods from a large volume of sample were required. However, these reported immunoseparation methods either required a large amount of immunomagnetic particles (up to 200 µg for each sample) [23] or used a small flow rate (0.15 µL/min) [29], or could only handle small volume of sample (50 µL) [33]. 

In this paper, we propose a fluidic device for immunomagnetic separation of pathogenic bacteria using self-assembled magnetic nanoparticle chains in a fluidic channel. A novel dot-array high gradient magnetic field was designed and developed using a permanent magnet and laminated sawtooth-shaped iron foils. Magnetic nanoparticle chains were formed on each tip of the sawteeth of the iron foils in the fluidic channel to improve the active contact between the immunomagnetic particles and target bacteria. *Salmonella typhimurium* with concentrations from 10^1^ to 10^4^ CFU/mL was used to verify the concept and evaluate the performance. 

## 2. Materials and Methods

### 2.1. Materials

The polyclonal antibodies against *salmonella* purchased from Abcam (ab69255, Cambridge, MA, USA) were used for specifically reacting with *salmonella typhimurium*. The carboxyl modified magnetic nanoparticles with an average diameter of 150 nm and a Fe concentration of 1 mg/mL were obtained from Ocean Nanotech (San Diego, CA, USA) for immunomagnetic separation. Phosphate-buffered saline solution (PBS, P5493, 10 times concentrated) from Sigma Aldrich (St. Louis, MO, USA) was 10 times diluted with deionized water to prepare the PBS solution (pH 7.4, 0.01 M). Tween-20 purchased from Amresco (Solon, OH, USA) was used for washing. Luria-Bertani (LB) medium (Aoboxing Biotech, Beijing, China) was used for bacteria culture. Bovine serum albumin (BSA) from EM Science (Gibbstown, NJ, USA) was used for blocking. All the solutions were prepared with deionized water produced by Advantage A10 from Millipore (Burlington, MA, USA). 

The silicone elastomer kit from Dow Corning (Sylgard 184, Auburn, MI, USA) was used for fabricating the fluidic chip. The Objet24 3D printer and the printing material (VeroWhite plus RGD835) from Stratasys (Eden Prairie, MN, USA) were used for fabricating the mold of the fluidic chip. 

### 2.2. Design and Fabrication of the Fluidic Chip

The proposed immunomagnetic separation device, as shown in Figure 1a, consisted of a fluidic chip with a separation chamber (width = 1 mm, height = 200 μm) for immune magnetic separation of target bacteria, a dot array magnetic field generator with laminated sawtooth-shaped iron foils and a permanent magnet for forming the magnetic nanoparticle chains, and anti-*salmonella* antibody modified magnetic nanoparticles with a diameter of 150 nm for reacting with the target bacteria.

The fluidic chip was fabricated based on 3D printing and surface plasma bonding. First, the mold of the fluidic chip was designed by Solidworks and fabricated using the Object24 3D printer with an accuracy of 100 μm in the X axis and Y axis and 28 μm in the Z axis. Then, the polydimethylsiloxane (PDMS) prepolymer and the curing agent were mixed at a ratio of 10:1 for 30 min and the mixture was cast into the mold after degassing in a vacuum for 20 min, followed by curing at 60 °C overnight. Finally, the PDMS channel was peeled from the mold and bonded with a clean glass treated using oxygen plasma (Harrick Plasma, Ithaca, NY, USA) to obtain the fluidic chip.

The dot-array magnetic field was generated using laminated sawtooth-shaped iron foils and a magnet. A long sawtooth-shaped iron foil with a thickness of 0.5 mm and a height of 5 mm, purchased from Harden (Shanghai, China), was first cut into 100 mm long pieces. Each piece had 100 regular triangle sawteeth with a side of 1 mm. Then, an iron foil holder was fabricated using the 3D printer to house 10 pieces of foils with a spacing of 1 mm. Finally, a strong NdFeB magnet (BZX084, grade: N42, surface field: 0.33 T, dimension: 4″ × 1/2″ × 1/4″), purchased from K&J Magnetics (Pipersville, PA, USA), was used to magnetize the iron foils to generate the local dot-array magnetic field. As shown in Figure 1b, under this proposed magnetic field, the magnetic nanoparticles could be formed into chains and uniformly distributed on top of the dot-array magnetic field in the fluidic channel.

### 2.3. Modification of the Immune Magnetic Nanoparticles

First, 150 μL of the streptavidin modified magnetic nanoparticles (diameter: 150 nm, Fe content: 1 mg/mL) were separated using a magnetic separator (SMS0206, Aibit, Wuxi, China) with a magnetic strength of 1.1 T for 2 min to remove the supernatant and resuspended in 500 μL of PBS (pH 7.4, 0.01 M), followed by incubating with 15 μL of the anti-*salmonella* polyclonal antibodies (2 mg/mL) for 45 min at 15 rpm to form the antibody modified magnetic nanoparticles (MNPs). Then, these immune MNPs were washed with 500 μL of PBS with 0.05% Tween 20 (PBST) twice to remove the redundant antibodies. Finally, the MNPs (0.15 mg/mL) were suspended in 1 mL of PBS with 1% BSA and stored at 4 °C for further use.

### 2.4. Immunomagnetic Separation of the Target Bacteria in the Fluidic Chip

Prior to test, the antibody modified magnetic nanoparticles were washed with 1 mL of the PBST and resuspended in 1 mL of PBS, and 1% BSA was used to block the fluidic channel for 20 min to avoid nonspecific adsorption, followed by washing with PBS (pH 7.4) for 2 min. First, 200 μL of the immune MNPs were injected into the fluidic channel using the peristaltic pump (ODM WX10, Longer, Baoding, China) with a flow rate of 250 μL/min, and the dot-array high gradient magnetic field was applied under the fluidic chip for 2 min, allowing the forming of the magnetic nanoparticle chains in the channel above each tip of the sawteeth. Then, 500 μL of the bacteria sample at a concentration of 10^4^ CFU/mL was continuous-flow injected into the channel at a flow rate of 50–250 μL/min, recycled for 45 min and washed with PBS. After the magnet was removed, the sample was flushed out of the channel with PBS at a flow rate of 1 mL/min and washed with PBST twice using the magnetic separator (SMS0206, Aibit, Wuxi, China). Finally, the bacteria were resuspended with 700 μL of PBS for culture plating to determine the amount of the separated bacteria. 

### 2.5. Preparation and Enumeration of the Target Bacteria

*Salmonella typhimurium* (ATCC14028) and *Escherichia coli* O157:H7 (ATCC43888) were used as target bacteria and non-target bacteria, respectively. Prior to use, they were both stored at −20 °C with 15% glycerol. For bacterial preparation, they were first grown in liquid Luria-Bertani (LB) medium at 37 °C for 12–16 h with shaking at 180 rpm after reviving. Then, the bacteria cultures were serially diluted with PBS to obtain the bacteria at concentrations from 10^1^ to 10^4^ CFU/mL.

For bacteria enumeration, the bacteria cultures were first diluted with PBS serially. Then, 100 μL of the diluent was surface plated on their respective selective agar plates and incubated at 37 °C for 22–24 h. Finally, the colonies were counted, and the bacteria were enumerated in colony forming units per milliliter (CFU/mL).

### 2.6. Calculation of Separation Efficiency

The separation efficiency (SE) of the proposed device is defined as the ratio of the amount of bacterial cells after separation to the amount before separation, and can be calculated using the following equation:(1)$$SE=\frac{N_{s}}{N_{C}}\times 100\%.$$
where $N_{s}$ is the amount of bacterial cells after separation (CFU/mL); and $N_{c}$ was the amount of bacterial cells before separation (CFU/mL).

## 3. Results and Discussions

### 3.1. Simulation of the Dot-Array Magnetic Field

The magnetic field plays an important role in the forming of the magnetic nanoparticle chains in the fluidic channel since the magnetic force applied on the magnetic nanoparticles is proportional to the magnetic density and gradient of the magnetic field. Thus, the finite element analysis software COMSOL (5.3a, Burlington, MA, USA). was used to simulate the dot-array magnetic field in this study. The magnetic density of the dot-array magnetic field is partially shown in Figure 2. The density of the magnetic field was about 0.2 T, and the gradient of the magnetic field was about 300 T/m.

### 3.2. Forming of the Magnetic Nanoparticle Chains

The size of the magnetic nanoparticles and the flow rate of the fluid also have great impacts on the forming of the magnetic nanoparticle chains. Since our previous study has shown that magnetic nanoparticles with a diameter from 100 nm to 200 nm have a higher separation efficiency of foodborne bacteria [9], nanoparticles with a size of 150 nm were used in this study. To verify the forming of magnetic particle chains in the fluidic chip, a fluorescence inverted microscope (Ti-E, Nikon, Tokyo, Japan) was used to observe the channel from a side view at different flow rates ranging from 10 μL/min to 350 μL/min. 200 μL of magnetic nanoparticles (Fe content: 0.15 mg/mL) suspended in PBS were injected to fill the fluidic chip using a precise syringe pump (KDS100, KD Scientific, Holliston, MA, USA). As shown in Figure 3a–h, when the flow rate was less than 150 μL/min, the magnetic particle chains basically retained a complete chain structure. When the flow rate increased to 250 μL/min, the magnetic nanoparticle chains were partially damaged and washed out, and when the flow rate was over 300 μL/min, the chains almost disappeared. In addition, microscopy was also used to observe the channel from the top. The same amount of magnetic particles were injected into the fluidic chip, followed by standing for 2 min to allow the forming of the chains. As shown in Figure 3i, the dot-array magnetic particle chains were successfully formed in the fluidic channel.

### 3.3. Immunomagnetic Separation of Salmonella Typhimurium in the Fluidic Chip

The immunomagnetic separation of *salmonella typhimurium* in PBS solution was performed in the proposed fluidic chip to test separation efficiency. Triplicate tests on 500 μL of *salmonella typhimurium* at a concentration of 10^4^ CFU/mL were conducted with three different flow rates (50 μL/min, 150 μL/min and 250 μL/min) for sample injection. As shown in Figure 4, a lower sample flow rate had a higher separation efficiency, which could be explained by the lower flow rate offering more immune reaction time for the antibodies on the chains to capture the target bacterial cells. In addition, when the flow rate was lower, the structure of the magnetic particle chains were more complete, thus leading to a better separation efficiency.

Three parallel tests were conducted for each concentration of *salmonella typhimurium* ranging from 1.6 × 10^1^ to 1.6 × 10^4^ CFU/mL in the PBS. As shown in Figure 5, the separation efficiency of this proposed device was ~80% for *salmonella typhimurium* with concentrations from 10^2^ to 10^4^ and ~65% for the concentration of 10^1^ CFU/mL, indicating that this fluidic device had good immunoseparation ability for the target bacteria with concentrations from 10^2^ to 10^4^ CFU/mL.

The specificity of immunomagnetic separation is also important to the detection of the target bacteria. Thus, the non-target bacteria, *E. coli* O157:H7, at the same concentration as the target bacteria, was used to evaluate the specificity. As shown in Figure 6, the separation efficiency of *E. coli* O157:H7 was only about 5%, which was much smaller than that of *salmonella typhimurium*. Furthermore, control experiments were also conducted by using magnetic nanoparticles without antibodies to separate *salmonella typhimurium* at a concentration of 10^4^ CFU/mL. The low separation efficiency of 6% indicated that this proposed immunomagnetic separation device had good specificity.

### 3.4. Comparsion of the Separation with and without Sawtooth-Shaped Iron Foils

To further verify the effect of the sawtooth-shaped iron foils on the separation of target bacteria, parallel tests on immunomagnetic separation using the fluidic chip with the sawtooth-shaped iron foils and the fluidic chip without the iron foils were conducted to separate 500 μL of *salmonella typhimurium* at a concentration of 10^4^ CFU/mL. The flow rate for the fluidic chip was 50 μL/min. The results are shown in Figure 7. The separation efficiency for the chip with the sawtooth-shaped iron foils was 77%, and much higher than that without the iron foils (36%). This could be attributed to the following reasons: (1) for the chip without the foils, the magnetic nanoparticles were aggregated at both ends of the channel since the block magnet has strong magnetic field intensity at both ends and weak intensity at the middle, resulting in a lower reaction efficiency of the magnetic nanoparticles with the target bacteria; (2) for the chip with the foils, the magnetic nanoparticles were more uniformly distributed above the tip of the sawteeth and formed vertical magnetic nanoparticle chains, resulting in a higher reaction efficiency of the nanoparticles with the bacteria. In addition, an interesting study on magnetic fluidized beds by Pereiro et al. [33] was used as a comparison with this proposed immune magnetic separation. The separation efficiency of target bacteria using the magnetic fluidized bed method was a little higher than that using this proposed method. However, this proposed device could handle a sample with a volume of 500 μL, which was 10 times larger than the magnetic fluidized bed method (50 μL).

## 4. Conclusions

In this paper, we successfully developed a fluidic device for the immunomagnetic separation of bacterial cells using magnetic nanoparticle chains, and evaluated its performance using *salmonella typhimurium* as a model. The forming of the magnetic nanoparticle chains was demonstrated to be effective for the separation of the target bacteria. Under the optimal conditions, the immunoseparation efficiency of the target bacteria was up to 80%. This proposed device has the potential to separate the target bacteria from samples with larger volume to further increase the sensitivity of biodetection, and can also be used with various biosensing platforms to provide low-cost and simple solutions for the rapid screening of foodborne pathogenic bacteria to enhance food safety.

## Figures and Tables

**Figure 1 micromachines-09-00624-f001:**
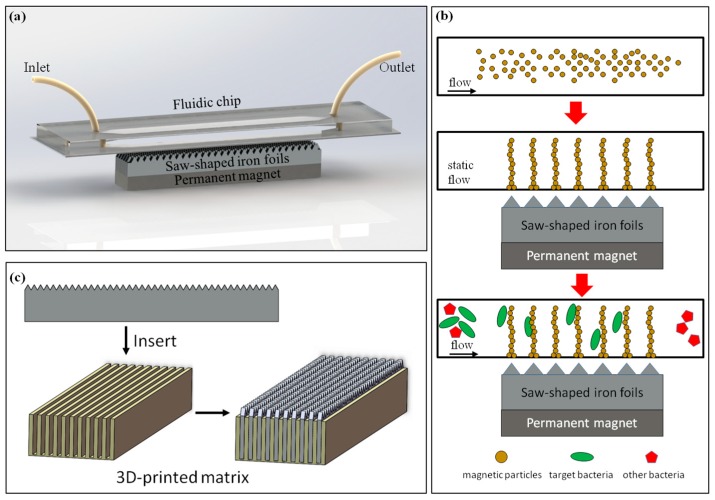
(**a**) Schematic view of the fluidic device for immunomagnetic separation; (**b**) the immunomagnetic separation of the target bacteria using nanoparticle chains in the channel; (**c**) the setup of the sawtooth-shaped iron foils.

**Figure 2 micromachines-09-00624-f002:**
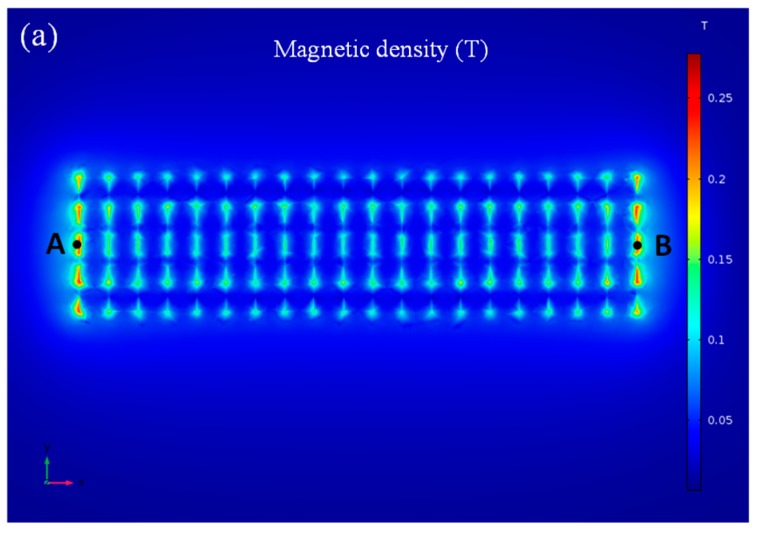
(**a**) Simulation of the magnetic field in the fluidic channel; (**b**) the magnetic density from point A to point B.

**Figure 3 micromachines-09-00624-f003:**
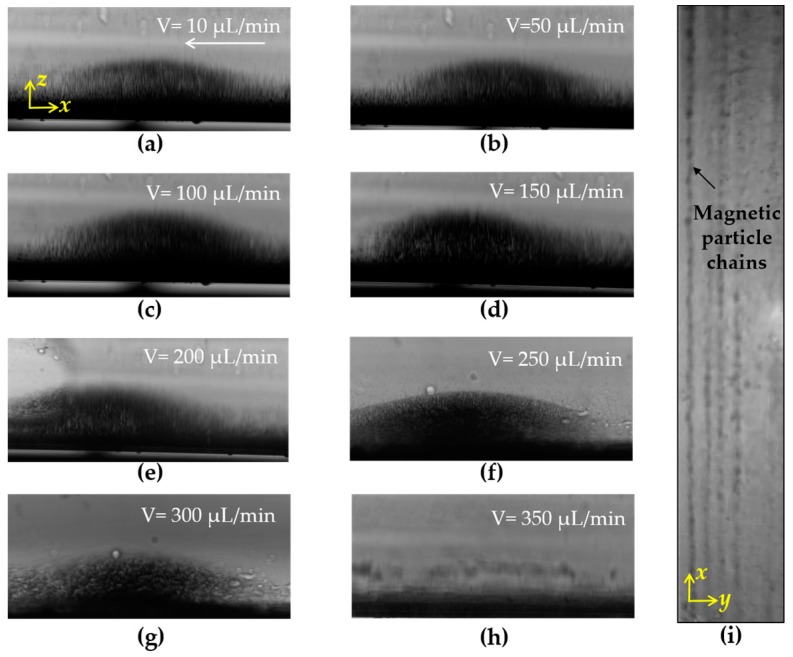
(**a**–**h**) Magnetic particle chains in the channel at different flow rates; (**i**) microscopic image of the dot-array magnetic particle chains.

**Figure 4 micromachines-09-00624-f004:**
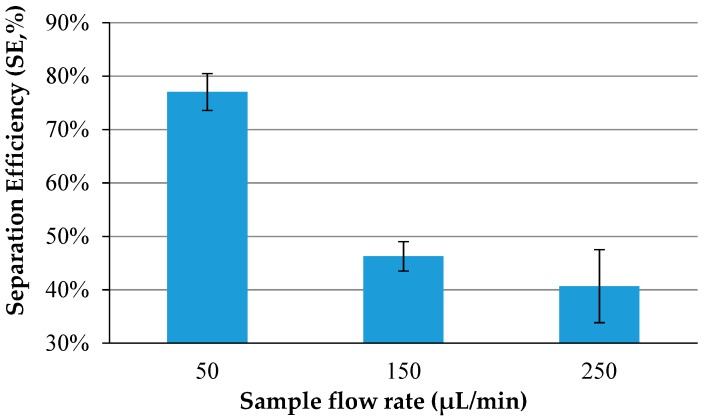
Separation efficiency of *salmonella typhimurium* at a concentration of 10^4^ CFU/mL the using proposed device with different flow rates (*N* = 3).

**Figure 5 micromachines-09-00624-f005:**
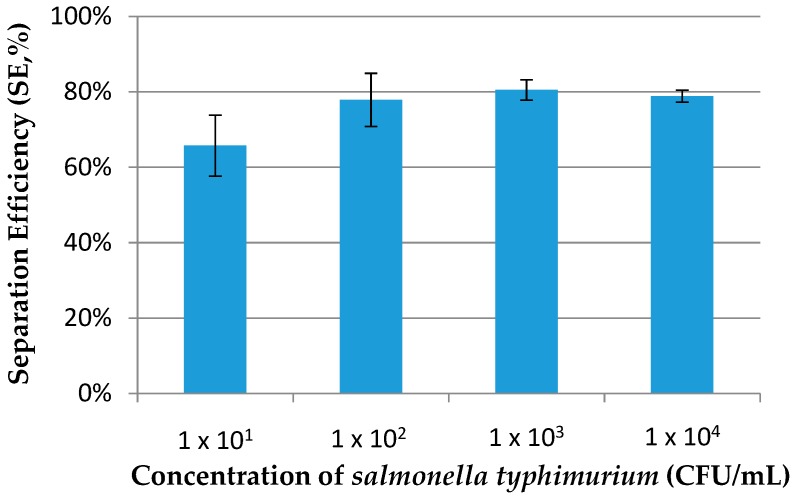
Separation efficiency of *salmonella typhimurium* at concentrations from 10^1^ to 10^4^ CFU/mL using the proposed device with a flow rate of 50 µL/min (*N* = 3).

**Figure 6 micromachines-09-00624-f006:**
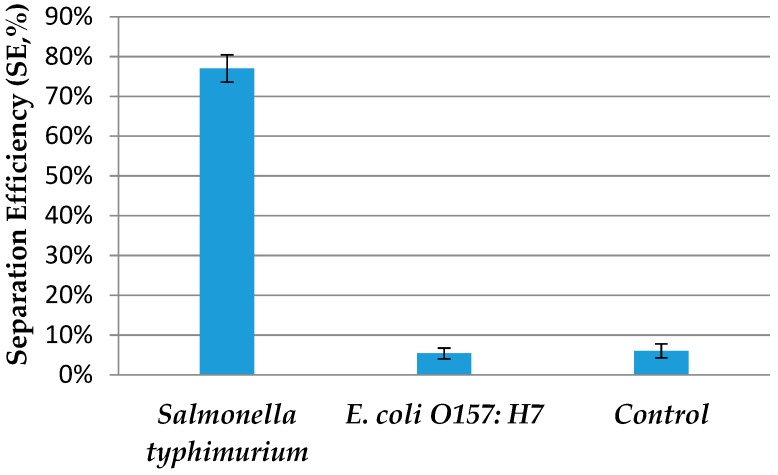
Separation efficiency of *salmonella typhimurium* and *E. coli* O157:H7 at a concentration of 10^4^ CFU/mL using the proposed device (*N* = 3).

**Figure 7 micromachines-09-00624-f007:**
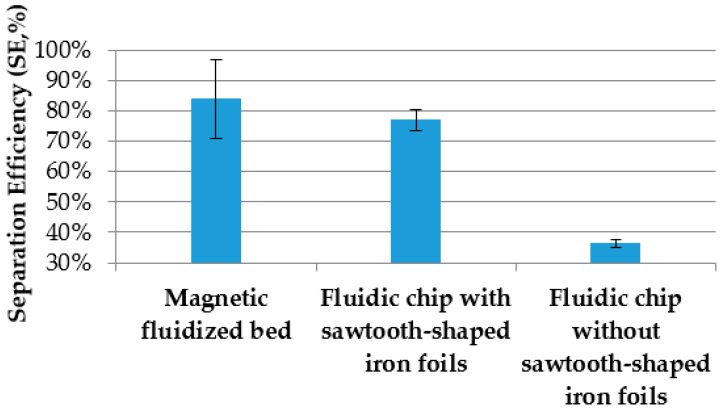
Separation efficiency of *salmonella typhimurium* at a concentration of 10^4^ CFU/mL using a magnetic fluidized bed and fluidic chip with and without the sawtooth-shaped iron foils (*N* = 3).

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
