# Peer review of "A Fluidic Device for Immunomagnetic Separation of Foodborne Bacteria Using Self-Assembled Magnetic Nanoparticle Chains"

_micromachines, 2018, doi:10.3390/mi9120624_

Reviewer 1 Report

This paper seems to be very practical for separation of foodborne bacterial cells using immunomegnetic methods. However, the review have some minor comments prior to publication in the Micromachines.

1.      In the test of forming of the nanoparticles chains (Fig. 3), Coud you show me more clear microscopic image data? The reviewer thinks that flourescence data of magnetic particle chain could be more helpful in this data.

2.     In the test of separation efficiency (Fig. 4), more lower flow rate (e.g. < 50 μl/min) can be applied for more efficient separation results. Do you have any data with lower flow rates?

Author Response

Please see our response to your comments in the document. Thank you very much.

Reviewer 2 Report

Cai et al have reported a microfluidic device for separation of salmonella spiked in PBS solutions at a concentration of 10,000 CFU/ml with an efficiency of 77% in 45 minutes. The chip consisted of a 3D printed fluidic channel mounted on a permanent magnet with a sawtooth-shaped iron foil in between them to enhance the magnetic field gradient in the channel, needed for magnetic nanoparticles self-assembly into immunomagnetic bacteria-capturing chains. This is an interesting method, but the paper needs improvement both in terms of more experiments needed to support the claims and more elaborate discussions of the results. I am providing detailed comments for the authors to improve the manuscript.

Major comments

1.     Please describe how novel the work is? This is not clear from the introduction section or the discussion of the results because the authors do not define a scientific or technological gap in the introduction section in the last two paragraphs, and also do not compare their work with existing continuous microfluidic separation methods (instead of the conventional centrifuge in Fig. 6). I recommend improving the introduction section and providing more discussion of results in comparison to the state of the art technologies.

2.     The flow rate chain breakup experiment is interesting, but the figure provided either does not convey the results very well, or some elaboration is needed to clarify the results. I recommend

a.      Adding a schematic of the imaging direction to Fig. 3.

b.     Demonstrating an experiment without any nanoparticles.

c.      Elaborating why fluorescent microscopy was need? Were the particles fluorescent?

d.     Reporting either the flow rates in Fig. 3 or velocities in Fig. 4 for better association of the two.

e.      Performing the experiment in PBS rather than DI water to resemble the biological tests more realistically.

3.     Bacteria concentration of 1,000 CFU/ml is extremely high when it comes to the application of interest in the manuscript which is food monitoring. More importantly, investigating the effect of bacteria and particle concentration on separation is an important parametric characterization needed. Eventually, it is extremely important to report on the limit of separation in this device.

4.     What was the effect of magnetic field strength on the results? Can the magnet assembly distance be changed to investigate this?

5.     One aspect of the test which is not clear is the washout process after bacteria capturing. Was there a washing step before the magnet was removed and the bacteria were recovered from the device? It is important to wash the uncaptured bacteria out before releasing the beads. Experiments are also needed to prove that the washing process efficiently cleaned the device before captured bacteria were released for plating. On the same topic, was there a need for nanoparticle separation from bacteria before plating?

6.     Can the simulation results in Fig. 2 be validated in any possible way, at least the 0.2 T magnetic field? Also, in lines 197-202, the authors discuss self-assembly of the nanoparticles, but this is not shown in the simulations and cannot be claimed in this section.

7.     It is beneficial to investigate the effect of lower flow rates in Fig. 4 to report if the efficiency can be further enhanced.

8.     Some control experiments with no nanoparticles and no antibodies are highly suggested.

9.     Please discuss how your technology differs from other microfluidic ones rather than a centrifuge. In the last figure, the centrifuge is better to be replaced with a comparable continuous assay. Moreover, the authors criticize the centrifuge in terms of continuous processing and limitation to 0.5 ml samples. They should acknowledge the fact that their technology will process a 0.5 ml sample in 100 min at 50ul/min which is relatively long, and at lower efficiency. 

Minor comments

1.     In line 78-79, the authors report the very large consumption of nanoparticles in previous works. Can they quantify the amounts and compare them to their method?

2.     3D printing results in some surface roughness in the channel. Is this roughness contributing to the results in any possible way such as trapping the nanoparticles or influencing the magnetic field? Can the authors verify their results in a smooth channel made by soft lithography?

3.     The sentence in lines 53-54 is vague. Please clarify

4.     There are minor spelling errors like in line 142, “cuted” should be “cut. 

Author Response

Please see our responses to your comments in the document. Thank you very much for your review on our manuscript.

Round  2

Reviewer 1 Report

This revised manuscript seems to be more acceptable in the present form.

Author Response

We would like to thank you again for your review on our manuscript with constructive suggestions for us to greatly improve our manuscript. All your comments have been carefully considered. The detailed point-by-point responses to your comments could be found in attached document.

Reviewer 2 Report

Thank you for addressing the comments. Please make sure that the presentation and formatting of the figures are professional and consistent across the paper. 

Author Response

(The authors gave the same response as above.)
